# Restoration of MHC-I on Tumor Cells by Fhit Transfection Promotes Immune Rejection and Acts as an Individualized Immunotherapeutic Vaccine

**DOI:** 10.3390/cancers12061563

**Published:** 2020-06-12

**Authors:** María Pulido, Virginia Chamorro, Irene Romero, Ignacio Algarra, Alba S-Montalvo, Antonia Collado, Federico Garrido, Angel M. Garcia-Lora

**Affiliations:** 1Servicio de Análisis Clínicos e Inmunología, UGC Laboratorio Clínico, Hospital Universitario Virgen de las Nieves, Av. de las Fuerzas Armadas 2, 18014 Granada, Spain; pulido_fresneda@hotmail.com (M.P.); vchamo26@hotmail.com (V.C.); albasanchezm1@correo.ugr.es (A.S.-M.); federico.garrido.sspa@juntadeandalucia.es (F.G.); 2Instituto de Investigación Biosanitaria ibs.GRANADA, 18012 Granada, Spain; 3UGC Laboratorios, Complejo Hospitalario de Jaén, 23007 Jaén, Spain; iren_romero@hotmail.com; 4Departamento de Ciencias de la Salud, Universidad de Jaén, 23071 Jaén, Spain; ialgarra@ujaen.es; 5Unidad de Biobanco, Hospital Universitario Virgen de las Nieves, 18014 Granada, Spain; antonia.collado.sspa@juntadeandalucia.es; 6Departamento de Bioquímica, Biología Molecular e Inmunología III, Universidad de Granada, 18071 Granada, Spain

**Keywords:** MHC-I restoration, Fhit, antitumor immunity, immune profile, cytotoxic T lymphocytes, immunotherapy, vaccine

## Abstract

The capacity of cytotoxic-T lymphocytes to recognize and destroy tumor cells depends on the surface expression by tumor cells of MHC class I molecules loaded with tumor antigen peptides. Loss of MHC-I expression is the most frequent mechanism by which tumor cells evade the immune response. The restoration of MHC-I expression in cancer cells is crucial to enhance their immune destruction, especially in response to cancer immunotherapy. Using mouse models, we recovered MHC-I expression in the MHC-I negative tumor cell lines and analyzed their oncological and immunological profile. Fhit gene transfection induces the restoration of MHC-I expression in highly oncogenic MHC-I-negative murine tumor cell lines and genes of the IFN-γ transduction signal pathway are involved. Fhit-transfected tumor cells proved highly immunogenic, being rejected by a T lymphocyte-mediated immune response. Strikingly, this immune rejection was more frequent in females than in males. The immune response generated protected hosts against the tumor growth of non-transfected cells and against other tumor cells in our murine tumor model. Finally, we also observed a direct correlation between FHIT expression and HLA-I surface expression in human breast tumors. Recovery of Fhit expression on MHC class I negative tumor cells may be a useful immunotherapeutic strategy and may even act as an individualized immunotherapeutic vaccine.

## 1. Introduction

The role of immune system in the control and destruction of tumor cells is well documented [1,2]. Moreover, treatments to activate the immune response against cancer cells or reverse tumor-promoted immunosuppression have produced tumor regression in cancer patients and have been approved for clinical application [3,4,5]. Among these, the use of antibodies against immune-checkpoints, vaccination, and adoptive cell therapies (ACT) have achieved promising results [6,7]. However, the success of many of these immunotherapies requires expression on the tumor cell surface of MHC class I (MHC-I) molecules loaded with specific tumor antigens [8,9,10]. Loss of this expression is the most frequent immune escape mechanism for cancer cells. Seven altered MHC-I phenotypes described in human tumors may allow cancer cells to be invisible to the immune system (immunoblindness) [11,12]; these phenotypes range from MHC-I allelic losses and MHC-I haplotype loss to the complete loss of expression of all six HLA class I molecules. The molecular mechanisms involved in these MHC-I alterations can be divided between: (a) irreversible mechanisms (“hard lesions”), including genetic changes in HLA class I heavy chains, antigen processing and presentation machinery (APM) components, or β_2_-microglobulin, or in other genes related to their expression; and (b) reversible mechanisms (“soft lesions”), including epigenetic, transcriptional, and posttranscriptional changes in these genes [13,14].

We previously described a new mechanism for the complete loss of MHC-I expression in our GR9 mouse tumor model system, derived from a fibrosarcoma produced by methylcholanthrene in BALB/c mice [15]. The B9 tumor clone has complete loss of MHC-I expression in baseline conditions due to a coordinated transcriptional downregulation of genes for several APM components, β_2_-microglobulin, and MHC-I heavy chains [16]. The B9 tumor clone also shows loss of the transcriptional expression of fragile histidine triad (Fhit) tumor suppressor gene. Stable transfection of the B9 tumor clone with the Fhit gene induced MHC-I surface expression, with recovery of the surface expression of H-2 K, D, and L molecules on the B9 tumor cells, which was mediated by transcriptional coordinated upregulation of APM, β_2_-microglobulin, and MHC-I heavy chain genes. Moreover, when Fhit expression was blocked in MHC-I positive metastatic clone MN4.5, a downregulation in MHC-I expression was observed. According to these results, the Fhit tumor suppressor gene may positively regulate MHC-I expression on murine tumor cells [15].

FHIT gene is a tumor suppressor located in the most common fragile site at human chromosome 3p14, FRA3B [17]. Loss of FHIT expression has been recorded in a high percentage (around 50%) of many human tumors, including those in breast, prostate, and lung, among others [18,19,20,21]. FHIT loss on cancer cells favors tumor progression and metastasization and is associated with a poor prognosis [22,23,24]. This gene has been found to inhibit the proliferation of tumor cells, induce their apoptosis, and increase invasion and their susceptibility to chemotherapeutic agents [25,26]; however, it has been also reported that FHIT does not act as a direct tumor suppressor gene [27,28,29].

The aims of the present study were: to determine the effect of transfecting Fhit in another MHC-I-negative tumor clone of the GR9 mouse tumor model, and to examine the effect of Fhit transfection and MHC-I expression restoration on the biological, oncogenic, and immunogenic characteristics of MHC-I negative tumor cells. Both B9 and B11 tumor clones are locally highly oncogenic [15,30], and our purpose was to explore modifications in their in vivo oncogenic and immunogenic potential caused by Fhit gene transfection and the restoration of MHC-I expression. Complete MHC-I cell surface downregulation or loss has been reported in around 50% of human breast tumors [31,32], and a further study aim was to explore the possible association between FHIT protein expression and MHC-I expression in human breast cancer cells.

## 2. Results

### 2.1. Stable Fhit Gene Transfection Recovers MHC-I Expression on B9 and B11 Tumor Cells

We determined the role of Fhit gene expression in MHC-I expression, oncogenicity, and immunogenicity in B9 and B11 tumor cell lines, both of which are MHC-I negative and recover expression of H-2 class I molecules K, D, and L after in vitro treatment with IFN-γ (Appendix A). B11 tumor cells present very weak expression of the H-2K and H-2D, and the three H-2 class I molecules are strong induced after treatment with IFN-γ. B9 tumor cells have no expression of MHC-I molecules, and after treatment with IFN-γ the induction of three H-2 class I molecules is lower. Furthermore, B11 tumor cells do not have the capacity to produce overt spontaneous metastases, while B9 tumor cells do. Previously, we showed that MHC-I expression on the B9 tumor clone was induced and recovered by stable transfection with the Fhit gene (Appendix A), promoting an increase to transcriptional and protein levels of APM and MHC-I heavy chains [15]. In the present study, stable transfection of B11 tumor clone with Fhit strongly induced MHC-I expression, recovering surface expression of the three H-2 class I molecules (Figure 1A). As controls, B9 and B11 tumor cells were transfected with the empty plasmid, finding no difference in MHC-I expression between control-transfected and non-transfected tumor cells. Because their results were identical in all assays, we only report here the data for non-transfected tumor cells. Fhit-transfected B11 tumor cells had increased transcriptional levels of the following genes: Fhit, H-2 class I heavy chains, β_2_-microglobulin, Tap-1, Tap2, LMP2, LMP10, Tapasin, calreticulin, Erap1, ERp57, PA28α, Ttp1, and Ttp2 (Figure 1B). All other analyzed APM genes showed no significant differences in their expression. Because these genes and MHC-I surface expression can be induced by in vitro IFN-γ treatment of these cells, we studied the effect of Fhit gene transfection on genes of the IFN-γ signal transduction pathway and its role in this induction. TB11-Fhit-transfected tumor cells showed a significant increase in the expression of IRF-1, Jak1, Ptpsh1, Stat1, Stat2, and Stat3 genes (Figure 1C). We also explored whether the induction by IFN-γ treatment of MHC-I expression on B11 tumor cells may be mediated by the induction of Fhit gene expression. Interestingly, results showed that Fhit gene expression was not induced by in vitro IFN-γ treatment of B11 tumor cells. Furthermore, MHC-I surface expression was induced on TB11-Fhit-transfected tumor cells by treatment with IFN-γ (Appendix A).

Fhit gene transfection did not modify the surface expression of non-classical MHC class I or classical MHC class II molecules on B11 tumor cells (Appendix A).

### 2.2. Fhit Gene Transfection Decreased the In Vitro Proliferation, Migration, and Invasion of Tumor Cells

In each of the following assays, we compared B9 with TB9-control and TB9-Fhit transfected tumor cells, and B11 with TB11-control and TB11-Fhit transfected tumor cells. Because identical results were obtained for the two tumor clones, we only report here on comparisons of B9 with TB9-control and TB9-Fhit transfected tumor cells. Furthermore, B9 and TB9-control tumor cells showing identical results in all assays, we only report here the data for non-transfected B9 tumor cells.

Fhit gene transfection did not modify the morphology of B9 tumor cells or their apoptotic or necrosis rates (Figure 2A; Appendix A). In vitro proliferation assays showed a lower growth rate for Fhit-transfected tumor cells than for non-transfected cells. At 48 h and 96 h, the growth rate was 60% lower in Fhit-transfected versus non-transfected cells (Figure 2B). In vitro migration and invasion assays showed that the migration rate was 46% lower and the invasion rate 26% lower for Fhit-transfected versus non-transfected B9 tumor cells (Figure 2C). RT-PCR analysis of the transcriptional expression of cell cycle genes showed a significantly increased expression of ccnA1, ccnB2, ccnD3, ccnE1, cdc25b, and E2F1 and a significantly decreased expression of ccnD2 in the Fhit-transfected versus non-transfected tumor cells (Figure 2D).

Comparison of the in vitro susceptibility of the cells to chemotherapeutic agents showed that Fhit-transfected tumor cells were more susceptible to paclitaxel and less susceptible to cisplatin in comparison to non-transfected tumor cells (Appendix A), while no difference was found in the susceptibility to doxorubicin or methotrexate. In our in vitro proliferation assays, Fhit-transfected tumor cells showed a very strong susceptibility to TNF-α and a partial susceptibility to IFN-γ, while TGF-β increased the proliferative rate of the non-transfected tumor cells but had no effect on this rate in the Fhit-transfected tumor cells (Appendix A).

### 2.3. Fhit-Transfected Tumor Cells Were Immune-Rejected In Vivo by Syngeneic Immunocompetent Hosts, with Differences between the Sexes

The in vitro changes in tumor cells resulting from Fhit transfection might modify their in vivo local oncogenicity. Therefore, assays were performed to compare in vivo the local growth rate among Fhit-transfected, control-transfected, and non-transfected tumor cells. Control-transfected and non-transfected B9 and B11 tumor cells were inoculated subcutaneously at different cell doses in male syngeneic immunocompetent mice, which all developed local primary tumors (Figure 3A; Appendix A). Interestingly, when Fhit-transfected tumor cells were inoculated in male mice, they were rejected in half of these (Figure 3A; Appendix A) and generated local primary tumors in the other half, although at a slower growth rate in comparison to B9 tumor cells (8 mm longer diameter at 60 days vs. 28 days, respectively). Different results were obtained in female mice: when non-transfected and control-transfected B11 and B9 tumor cells were inoculated at different cell doses, they produced local primary tumors in 80% of the animals, at a similar growth rate to that in the males. Strikingly, inoculated Fhit-transfected tumor cells were rejected by all of the female mice (Figure 3A; Appendix A). These results reveal a strong rejection of Fhit-transfected tumor cells, with a significant difference in frequency between the sexes, 50% in males and 100% in females (*** *p* < 0.001, Fisher test) (Figure 3A; Appendix A). Minor male histocompatibility antigens on the Y chromosome cannot explain these sex-related differences in rejection, because cytogenetic analysis of B9 and B11 revealed that both cell lines are X chromosome monochromatic and lack a Y chromosome (Appendix A).

Given that Fhit-transfected tumor cells recovered their MHC-I expression, we then explored whether the in vivo rejection of these cells involved the immune system, mainly T lymphocytes. Fhit-transfected B9 and B11 tumor cells were inoculated in female and male nude mice lacking T lymphocytes and grew locally in all animals (Figure 3B; Appendix A). According to these results, T lymphocytes were responsible for the high immunogenicity of Fhit-transfected tumor cells in immunocompetent mice. The specific lymphocyte subpopulations involved were investigated by depleting immunocompetent male and female mice with a weekly intraperitoneal injection of anti-CD4 or anti CD8 specific antibodies before injecting them with Fhit-transfected tumor cells. Local primary tumors appeared in all CD8+T lymphocyte (CTL)-depleted immunocompetent mice, indicating that these lymphocytes were responsible for the immune rejection of Fhit-transfected tumor cells (Figure 3B; Appendix A). The tumor growth rate in these immunodepleted hosts was very similar to that observed with Fhit-transfected cells in male immunocompetent hosts, with the longest diameter of the primary tumor reaching 8 mm in 59 days. In other assays, immunocompetent mice that were CTL-depleted 60 days after the injection of Fhit-transfected tumor cells showed no local tumor growth, indicating that the CTLs destroy and eradicate Fhit-transfected tumor cells.

### 2.4. Changes in Immune Cell Subpopulations Produced by Fhit-Transfected Tumor Cells in Female and Male Immunocompetent Mice

Weekly flow cytometry analyses of spleen leukocyte subpopulations in female and male mice showed statistically significant differences (*p* < 0.05) between female mice inoculated with TB9-Fhit tumor cells in comparison to PBS-inoculated controls at 14 dpi, with increased B lymphocytes (51.5 vs. 42.5%) and decreased T lymphocytes (43.6 vs. 51.6 %) (Table 1). Interestingly, the female mice then showed a strong increase in T lymphocytes of up to 56.6% at 21 dpi, corresponding to an increase in T-cytotoxic lymphocytes (CTLs) (27%). This increase was higher at 28 days dpi after the reinjection of TB9-Fhit tumor cells at 21 dpi, reaching 61% T lymphocytes, with increases in both T-helper lymphocytes and CTLs (26.6 and 34.3, respectively) (Table 1). Different results were observed for the male hosts, detecting a slight increase in B lymphocytes at 14 and 21 dpi (51.6 and 50.2 vs. 47.7%) and only observing an increase in T lymphocytes at 28 dpi (49.4 vs. 44%) after reinjection of TB9-Fhit cells at 21 dpi, corresponding to a slight increase in T-helper lymphocytes and CTLs (Table 1). Female but not male mice showed a significant increase in cell-surface expression of T lymphocyte activation markers at 14 dpi, which was maintained until 28 dpi (Appendix A). We highlight the major differences in spleen lymphocyte subpopulations between female and male mice.

### 2.5. Changes in the Secretion of Chemokines/Cytokines by the Tumor Cells and Their Presence in the Plasma of Mice Inoculated with Fhit-Transfected Tumor Cells

Measurements of 36 different chemokines in the culture medium revealed a significant decrease in the secretion of CCL5, CCL8, CXCL1, CXCL2, CXCL5, and TGFβ1 and a significant increase in TGFβ2 secretion by TB9-Fhit transfected tumor cells (Figure 4A). All of these changes were at the transcriptional level, and mRNA-NGS assays revealed the same results for mRNA levels of these chemokines. All other analyzed chemokines/cytokines showed no significant differences in their expression.

Plasma chemokine levels were also compared between female or male mice inoculated with TB9-Fhit transfected tumor cells and PBS-inoculated mice (controls). Both sexes showed: a significant increase in CXCL11 and IL-16 at 56 days post-injection (dpi); a transient increase in CXCL12, CXCL13, and CXCL16 at 42 dpi, with their return to baseline levels at 56 days dpi; and an increase in CCL24 at 42- and 56-days dpi (Figure 4B). Sex differences were found at 42- and 56-days dpi, observing an increase in IL-1β for females and IL-10 for males at both time points and a transient increase in CXCL10 for females and CXCL5 for males at 42 days. A difference was found between male mice with and without primary tumor differed in IL-10 and IL-16 levels, which were higher in those with tumor, and in CXCL5, CXCL11, CXCL13, CXCL16, and CCL24 levels, which were higher in those without tumor (Figure 4B). All other analyzed chemokines/cytokines showed no significant differences in their expression.

### 2.6. Fhit-Transfected Tumor Cells Act as Individualized Immunotherapeutic Vaccine against GR9 Tumor Cell Clones

In assays using Fhit-transfected tumor cells as immunotherapeutic vaccine, female immunocompetent mice were inoculated with transfected TB9- or TB11-Fhit tumor cells. At 60 dpi, when these tumor cells were immune rejected, the hosts were inoculated with the corresponding B9 or B11 tumor cell clone. Notably, B9 and B11 tumor cells were totally immune rejected in all mice (Figure 5A). In similar experiments using other tumor cell clones of the GR9 tumor system with different MHC-I expressions (A7-MHC+++, B7-MHC++, C5-MHC+), all of the tumor clones were totally immune rejected by hosts previously inoculated with Fhit-transfected tumor cells (Figure 5B). In a similar immune protection assay against GR9 bulk tumor cells (original primary tumor composed of all tumor cell clones), GR9 tumor cells were totally immune rejected by the hosts (Figure 5B). According to these results, the hosts were fully protected by Fhit-transfected tumor cells against all tumor cell clones derived from the GR9 tumor system and even against GR9 bulk tumor cells. To determine the immune response involved in the rejection, female immunocompetent mice injected with Fhit-transfected B9/B11 tumor cell lines were depleted of CD4- or CD8-T-cells starting at 60 dpi, and a subsequent challenge with B9/B11 tumor cells was performed. The results showed that primary tumors generated from B9 or B11 tumor cells only grew in all animals depleted of CD8-T-cells, indicating that cytotoxic T-lymphocytes are involved in the immune rejection of untransfected tumor cells. In other assays, we injected B9/B11 tumor cells in immunocompetent mice, when the diameter of the tumor reached 10 mm it was removed, and then the animals were injected with the Fhit-transfected tumor cells. The results showed that the primary tumors did not re-grow (preventing re-growth that occurred in 30% of immunocompetent animals non-inoculated with Fhit-transfected tumor cells), and in the case of B9, no spontaneous metastases were generated.

The possible extension of this immune protection to other murine tumors was examined by inoculating 4T1 mammary carcinoma cells or CT26 colon carcinoma cells in hosts previously inoculated with Fhit-transfected tumor cells. The growth of local primary tumors derived from 4T1 or CT26 was observed in all mice (Figure 5C). In consequence, the immune response generated in the hosts by TB9- or TB11-Fhit transfected tumor cells generated individualized immune protection against tumor cells of the GR9 murine tumor system.

### 2.7. Loss of FHIT Expression in Human Breast Tumors Is Associated with Complete Loss or Downregulation of MHC-I Expression

Loss of Fhit expression on murine tumor cells derived from the GR9 tumor system produced complete loss of MHC-I expression, and Fhit gene transfection promoted the recovery of this expression on these tumor cells. We next examined whether this association between FHIT expression and MHC-I expression can be observed in human tumor cells, specifically choosing breast tumors because of their frequent complete loss of HLA-I cell surface expression (in ~50% of cases) [31,32]. Table 2 exhibits results of the immunohistochemical study of FHIT and HLA-I expression in samples from 48 human breast tumors: intense FHIT expression was observed in 16 tumors, 15 with intense HLA-I expression and 1 with moderate HLA-I expression (Figure 6A; Table 2); moderate FHIT expression was observed in 6 tumors, 1 with intense, 4 with moderate, and 1 with negative HLA-I expression (Figure 6B; Table 2); weak FHIT expression was observed in 15 tumors, 9 with weak and 6 with negative HLA-I expression (Figure 6C; Table 2); and negative Fhit expression was observed in 11 tumors, all with negative HLA-I expression (Figure 6D; Table 2). Tumor sample 32-07 is the only one with clearly discordant FHIT and HLA-I expressions (partial loss of FHIT and complete loss of MHC-I expression); therefore, this complete loss of HLA-I expression may have resulted from a molecular mechanism in which FHIT is not involved. Among the other 47 tumor samples, similar results were obtained for FHIT and HLA-I protein expressions in 39 tumor samples and only a very small difference in 8 (Table 2). All samples with loss of FHIT expression except one (*n* = 31 tumors) also showed loss of HLA-I surface expression. Our analysis therefore revealed a significant direct correlation between FHIT and MHC-I protein expression in human breast cancer (Spearman’s Rs = 0.925, *p* < 0.001).

## 3. Discussion

The restoration of MHC-I expression in cancer cells is crucial to enhance their immune recognition and destruction by CTLs [33]. In this study, the loss of Fhit gene expression was responsible for the coordinated transcriptional downregulation of APM and MHC-I heavy chain genes and the resulting complete loss of MHC-I surface expression in murine tumor cells. Fhit transfection completely reversed this process, increasing at transcriptional level genes of the IFN-γ signal transduction pathway. Because IFN-γ treatment of tumor cells has similar effects at transcriptional level, inducing MHC-I expression in vitro, we hypothesized that the induction of MHC-I expression by Fhit might be mediated by the following signal pathway: Fhit → IFN-γ-pathway genes → APM/MHC genes → MHC-I cell surface expression. However, Fhit is not involved in the induction of MHC-I surface expression produced by in vitro treatment with IFN-γ, which did not induce Fhit expression and which did induce MHC-I surface expression on Fhit-transfected tumor cells. Fhit is not a transcription factor and is located in the cytoplasm, so increased transcriptional level of these genes may be due to stabilization, interaction, or binding of Fhit with other proteins. It has been widely reported that Fhit modifies the expression of various signal transduction pathways through these processes of interaction forming complexes with other proteins [34,35,36,37]. Another possibility is that the loss of Fhit expression promotes the expression of a transcriptional repressor of these genes and that the reexpression of Fhit causes its binding to this repressor protein.

The precise mechanisms underlying tumor suppression by Fhit remain unclear. We found that Fhit-transfection of these tumor cell lines reduced their in vitro proliferation, migration, and invasion but did not produce apoptosis or necrosis. According to previous findings of our group, the increase in MHC-I expression induced by Fhit transfection may be directly responsible for these biological modifications observed in vitro [38]. The reduction in in vitro proliferation, migration, and invasion might therefore be thought responsible for the in vivo finding of no growth by Fhit-transfected tumor cells in immunocompetent mice. However, this possibility was ruled out because Fhit- transfected tumor cells grew and generated local primary tumors in nude mice and in CD8+ T lymphocyte-depleted immunocompetent mice. The loss of MHC-I expression on B9 and B11 tumor cells (immunoblindness) allowed tumor progression, whereas restoration of MHC-I expression by Fhit transfection promoted tumor visibility and destruction by a CD8+T lymphocyte-mediated immune response. Taken together, these data provide the first evidence of an important immune component in the action of Fhit as tumor suppressor gene. They also show that the immune system may be capable of completely eradicating cancer cells.

We highlight the major difference in immune rejection frequency between female (100%) and male (50%) animals. The sexes also differed in the host immune response (immune subpopulations and chemokine secretions) generated by Fhit-transfected tumor cells. A strong increase in T lymphocyte activation markers was found in female hosts at 14 dpi, promoting a subsequent increase in T lymphocytes, mainly CTLs, whereas no T lymphocyte activation or proliferation was observed in males. With regard to plasma chemokine levels, IL-1β was increased in females and IL-10 was increased in males, more markedly in the males with tumor. This difference in antitumor immune response between males and females suggests that the effectiveness of different anticancer immunotherapies or vaccines may possibly depend on the sex of the host [39,40,41,42]. In this regard, it was recently reported that vaccine efficacy and protection against influenza virus is greater in female than male mice [43]. Interestingly, the Fhit-transfected tumor cells acted as an immunotherapeutic antitumor vaccine. All female mice injected with these cells gained immune protection against the progression of cancer generated from the same (non-transfected) tumor cells, from another tumor clone derived from GR9 tumor model system, and even from GR9 bulk tumor cells. Tumor clones in the GR9 clone system vary in their MHC-I surface expression under baseline conditions, but these MHC-I alterations are reversible “soft lesions”, and MHC-I expression can be induced by IFN-γ treatment [44]. All of these tumor cells might recover MHC-I surface expression in vivo, and metastases from B11 or B9 tumor clone were found to be MHC-I-positive in immunodeficient hosts [30,45], and could therefore be recognized and destroyed in the immune response generated by Fhit-transfected tumor cells. Moreover, primary tumor and dormant metastases derived from B11 cells in immunocompetent mice recovered H-2 class I expression [15,30]. We previously reported that GR9, A7, and B7 tumor cells (all MHC-I positive) grew in vivo and generated immunosuppression in hosts [44,46]. The prior inoculation of Fhit-transfected tumor cells might reverse this immunosuppression.

Our results reveal for the first time that FHIT expression and MHC-I expression are significantly associated in human breast tumors (*p* < 0.001). All breast tumors with decreased FHIT protein expression also showed loss of HLA-I surface expression; moreover, 33% of the breast tumors had no loss of HLA-I surface expression or FHIT protein expression. In Tasmanian Devils, downregulation of MHC-I due to transcriptional downregulation of APM and MHC-I genes, and transcriptional downregulation of Fhit and its regulator mir-29b have been reported in DFTD tumor cells [47,48]. Hence, the positive regulation of MHC-I by Fhit may extend to other species. Various viruses (e.g., HPV in cervical cancer) cause MHC-I downregulation on tumor cells [49]. The fragile site in which FHIT gene is localized within human chromosome 3p14 region also contains an HPV virus integration site [50]. We therefore speculate that HPV integration might cause FHIT expression loss and consequently MHC-I downregulation on cervical cancer cells.

The NLR Family CARD Domain Containing 5 (NLRC5) gene has been described as a transcriptional coactivator of MHC-I baseline expression alongside other regulation mechanisms [51]. The expression of NLRC5 was not increased in TB11- and TB9-Fhit-transfected cells (unpublished observation). According to these observations, the regulation of MHC class I expression on tumor cells by Fhit is not mediated by NLRC5 gene expression. In addition, elevated MHC-I expression on various immune cells and normal T-cell development have been observed in NLRC5-knockout mice [52]. In human breast tumors, the loss of NLRC5 expression is highly infrequent, whereas around half of cases show complete loss of MHC-I expression [53].

## 4. Materials and Methods

### 4.1. Animals

Eight-week-old male or female BALB/c mice (Charles River Laboratories, Barcelona, Spain) were used in experiments. This study was carried out in accordance with the recommendations of European Community Directive 2010/63/EU and Spanish law (Real Decreto 53/2013) on the use of laboratory animals, and their housing and the experimental procedures were approved by the Junta de Andalucía animal care committee and adhered to animal welfare guidelines of the National Committee for Animal Experiments. The animals were anesthetized with 0.04 mL diazepam (Valium, Roche, Basel, Switzerland) and 0.1 mL ketamine (Ketolar, Pfizer, Kent, UK) When clear signs of disease were observed, the animals were anesthetized and euthanized by cervical dislocation, followed by complete necropsy.

### 4.2. Cell Lines and Stimulation

The GR9 cell line is derived from a mouse fibrosarcoma induced by methylcholanthrene in BALB/c mice and has been extensively characterized in our laboratory. It is composed of cell clones with different H-2 class I expression patterns. GR9-B9 and -B11 are two different clones obtained by limited dilution method from the GR9 tumor cell line and were recloned by capturing individual cells under phase contrast microscopy. CT26 is a BALB/c mouse-derived colon carcinoma cell line, and 4T1 is a BALB/c mouse-derived mammary carcinoma cell line; both were obtained from ATCC. All cell lines were characterized by PCR assay using short tandem repeats and were regularly tested for MHC-I surface expression. Cell lines were maintained in Dulbecco’s medium supplemented with 10% fetal bovine serum (Life Technologies, Thermo Fischer Scientific, Carsbald, CA, USA), 2 mM glutamine, and antibiotics. In some experiments, cell lines were treated with 50 U/mL IFN-γ or with different concentrations of TNF-α, TGF-β (Miltenyi-Biotech, Bergisch Gladbach, Germany), methotrexate, cisplatin, paclitaxel, and doxorubicin (Sigma-Aldrich, St. Luis, MO, USA) for 48 h.

### 4.3. Cloning of Fhit cDNA and Transfection into B11 Cells

Fhit cDNA was cloned from murine cells. Briefly, the total RNA was extracted using the RNeasy Mini kit (Qiagen, Valencia, CA, USA), and first-strand cDNA was synthesized using oligo (dT) primer and Superscript III Reverse Transcriptase (Invitrogen, Paisley, UK). The PCR was done using 20 pmol primers: Fhit-5′Fw- 5′-CACCATGTCATTTAGATTTGGCCAA-3′; and Fhit-3′-Bv- 5′-TCAGGCCTGAAAGTAGACCCG-3′. The Fhit amplification protocol consisted of denaturation for 1 min at 94 °C, annealing for 1 min at 65 °C, and extension for 2 min at 72 °C for a total of 30 cycles. The PCR product was purified and cloned using the pcDNA3.3/TA Expression kit (Invitrogen, Thermo Fischer Scientific, Carsbald, CA, USA), and the plasmid obtained was named pcDNA-Fhit. The correct sequence of the cloned Fhit cDNA was confirmed using an ABI PRISM 310 automated sequencer. The empty vector was used as control vector (pcDNA-control). The vectors were used to transform TOP10 bacteria according to the manufacturer’s instructions. The transformed bacteria colonies were selected in a medium containing 50 µg/mL ampicillin and expanded for plasmid DNA preparation. The plasmid DNA was extracted and purified using QIAprep plasmid spin columns (Qiagen). Next, B9 or B11 cells (1.5 × 10^5^/mL) were cultured in six-well plates at 37 °C in a 5% CO_2_ atmosphere and then transfected with 1 µg of pcDNA-Fhit plasmid or pcDNA-control using Lipofectamine 3000 reagent (Invitrogen) according to the manufacturer’s instructions. Transient transfection assays were performed for 48 h. Stable transfected cells were selected by using 1 mg/mL Geneticin (G-418, Life Technologies, Thermo Fischer Scientific, Carsbald, CA, USA), and viable cells were cloned by limited dilution, expanded, and cultured with continuous selection using Geneticin (400 µg/mL); they were designated TB9-Fhit and TB11-Fhit cells or TB9-control and TB11-control cells, respectively. The stable transfection of the Fhit was confirmed by reverse transcription-PCR (RT-PCR) and Western-blot.

### 4.4. MHC Class I and MHC Class II Cell Surface Expression

MHC class I cell surface expression was analyzed by indirect immunofluorescence using FACS (FACSCanto; Becton Dickinson, Franklin Lanes, NJ, USA) according to a standard protocol. Briefly, 5 × 10^5^ cells were washed twice with PBS and incubated for 30 min at 4 °C with the primary antibodies anti-H-2 K^d^ (K9.18), anti-H-2 D^d^ (34.5.8), anti H-2 L^d^ (28.14.8 and 30.5.7) (ATCC), anti H-2 Qa1 (6A8.6F10), anti H-2 Qa2, and anti H-2 I-A/I-E (M5/114) (Miltenyi-Biotec, Bergisch Gladbach, Germany). The secondary fluorescein isothiocyanate (FITC)-conjugate antibody (anti-mouse FITC IgG/Fab, Sigma-Aldrich, St. Luis, MO, USA) was used at 1:120 dilution for 30 min at 4 °C in the dark. Isotype-matched non-immune mouse IgG and cells labeled with the fluorescein-conjugated antibody alone served as controls. A minimum of 1 × 10^4^ cells were analyzed with CellQuest Pro software. All cell lines were studied in baseline conditions and after IFN-γ treatment.

### 4.5. RT and Quantitative Real-Time PCR

A mRNA isolation kit (Miltenyi-Biotech, Bergisch Gladbach, Germany) was used to extract mRNA from tumor cell lines. First-strand cDNA was synthesized with 100 ng mRNA using a High Capacity Reverse Transcription Kit (Applied Biosystems, Carlsbad, CA, USA) in a total volume of 20 µL. These cDNAs were diluted to a final volume of 100 µL. Real-time quantitative PCR analyses were performed in the 7500 Fast System (Applied Biosystems), performing PCR reactions in quadruplicate and expressing the values obtained as means ± standard deviation (SD). Quantitative PCR was performed with the Power SYBR Green Master mix (Applied Biosystems). Appendix A lists the selected genes, primers, and amplicon sizes. Data were normalized using β-actin and GAPDH as housekeeping genes. PCR conditions were 40 cycles of 15 s of denaturation at 95 °C and 60 s at 60 °C.

### 4.6. Cell Proliferation, Migration, and Invasion Assays

A total of 3 × 10^5^ or 1 × 10^6^ tumor cells were seeded in a T-25 or T- 75 cell culture flask, and total cell numbers were counted daily after 1–4 days by Trypan Blue exclusion method using a hemocytometer. Two investigators independently counted the cells. The proliferation index was calculated: final cell number/initial cell number. Migration and invasion assays were performed using Boyden Chambers containing polycarbonate filters with 8 µm pore size (Corning Life Sciences, New York, NY, USA). For the migration assays, the filters were coated with collagen; and for the invasion assays, growth factor-reduced Matrigel matrix-coated transwell chambers (Corning) were used. Tumor cell lines were seeded at a density of 2 × 10^4^ cells per well in serum-free medium in the upper chamber. The lower chamber was also filled with the serum-free medium. After 24 h serum starvation, the serum-containing culture medium was added in the lower chamber. After 20 h incubation at 37 °C in a 5% CO_2_ humidified incubator, membranes from the upper chamber were scrubbed using a medium-wetted cotton swab, and cells adhering to the underside of the Transwell membrane were fixed with 20% methanol and stained with 0.1% crystal violet. Forty areas per membrane sample were randomly selected under an inverted light microscope at 200× magnification, and the cells were counted. Each assay was repeated at least three times and was performed in duplicate

### 4.7. In Vitro Assessment of Apoptosis

Annexin V-FITC/PI staining method was used to identify and quantify apoptotic cells. Cells (3 × 10^5^/mL) were treated according to the manufacturer’s instruction of ANNEX300F Kit (Bio-Rad, Hercules, CA, USA). Cells were stained with annexin V-FITC at room temperature for 10 min in the dark, washed and then stained with PI. FITC/PI fluorescence intensity was measured by flow cytometry to differentiate between viable (annexin V-negative and PI-negative), early apoptotic (annexin V-positive, PI-negative), and late apoptotic/necrotic (annexin V-positive and PI-positive) cells.

### 4.8. In Vivo Local Tumor Growth Assay

In the in vivo oncogenicity assays, two cell doses, 6.25 × 10^5^ and 12.5 × 10^5^ cells, of each cell clone were subcutaneously inoculated into the footpad in groups of 10 mice. Two groups of mice were analyzed for each cell dose. The growth of local tumors was recorded three times/week in all animals, measuring the largest diameter of each tumor with electronic calipers.

### 4.9. Spectral Kariotyping

Cells were incubated in 200 ng/mL Colcemid (KaryoMAX Colcemid solution; Gibco) for 4 h and then incubated in 75 mM KCl. Chromosomes were subsequently fixed in 3:1 methanol:acetic acid and dropped onto glass slides. Metaphase spreads were incubated with mouse SkyPaint kit probe (Applied Spectral Imaging, Carlsbad, CA, USA) according to the manufacturer’s protocol. Image acquisition was performed using a SpectraCube system, and analyses were accomplished using the SkyView imaging software (Applied Spectral Imaging). A minimum of 20 metaphases were analyzed for each cell line.

### 4.10. Determination of Cytokine Levels

A Bio-Plex Pro™ Mouse Chemokine Panel 33-Plex, a Bio-Plex Pro™ Mouse MCP-2 single-plex, and Bio-Plex Pro™ Mouse TGF-β 3-Plex assays (Bio-Rad) were employed to profile the concentration of mouse cytokines and chemokines in cell culture supernatants and mice plasma according to the manufacturer’s instruction. Analysis of each sample was performed in duplicate and run on two separate occasions. Cell lines were seeded to 10^6^ cells and then, after incubation for 24 h, cells were counted and culture supernatants (50 µL) were transferred in duplicate to plates precoated with cytokine-specific antibodies conjugated with different color-coded beads, and the plates were incubated for 1 h, washed, and then sequentially incubated with 50 µL of biotinylated cytokine-specific detection antibodies and streptavidin-phycoerythrin conjugate. Positive and negative quality controls were included in duplicate in each assay. Fluorescence was recorded using a Bioplex 200 instrument, and cytokine/chemokine concentrations were calculated with Bio-Plex Manager software by using a standard curve derived from recombinant cytokines. Results were normalized to million cells/mL. Plasma from the blood of each mouse inoculated with tumor cells or PBS (controls) (*n* = 8 mice per group; all assays were repeated twice) was collected at different time periods, and they were processed as described above.

### 4.11. Isolation of Splenic Leukocytes

Spleens were excised and collected weekly from 14 to 56 days post-inoculation (dpi) of tumor cells in groups of 8 mice. All assays were repeated twice. Another assay was performed reinjecting the mice at 21 dpi again with Fhit transfected tumor cells, and at 28 days the spleens were excised. They were dissociated into single-cell suspensions using a gentle MACS Dissociator (Miltenyi Biotech, Bergisch Gladbach, Germany). Each spleen was dissociated by loading a C tube on the machine with a volume of buffer (0.01 mol/L phosphate-buffered saline [PBS], 0.5% bovine serum albumin [BSA], and 2 mmol/L ethylenediaminetetraacetic acid [EDTA]), selecting an installed program. After completion of the program, whole-cell suspensions were centrifuged at 300× g at room temperature for 10 min. Then, the suspensions were collected and filtered through a 70-µm-pore-size nylon cell strainer to remove clumps and generate single-cell suspensions. Red blood cells were lysed with ACK lysing buffer (Gibco) for 5 min and then washed twice in PBS. Viable cells were counted and used for the antibody staining reaction.

### 4.12. Flow Cytometry Analysis of Immune Cell Subsets

The following labeled antibodies (Miltenyi Biotec, Bergisch Gladbach, Germany) were used for the direct immunofluorescence study: CD45-PerCP, CD45-FITC, CD3-FITC, CD3-PE, CD19-PE, CD4-PerCP-Vio700, CD4-FITC-Viobright, CD8-PE, CD8-PerCP-Vio700, CD25FITC-Viobright, FoxP3-PE, CD49b-PE, MHC class II-PerCP-Vio700, CD11c-PE, CD11b-FITC-Viobright, F4/80-PE, CD134-PE, CD154-PE, CD137-PE, CD69-PE, CD178-PE, CD44-PE, CD95-FITC-Viobright, CD62L-FITC, TCRαβ-PE, and TCRγδ-PE (Milyenyi-Biotec, Bergisch Gladbach, Germany). Isotype-matched non-immune mouse IgGs conjugated with FITC, PE, or APC and unstained cells served as controls. FcR Blocking Reagent was used to block unwanted binding of antibodies to mouse cells expressing Fc receptors. Briefly, 5 × 10^5^ cells were washed twice with PBS and incubated for 10 min at 4 °C in the dark with the primary antibodies. For the determination of Treg cells, cells were incubated for 10 min at 4 °C in the dark with anti-CD4 and anti-CD25 antibodies after permeabilization for 30 min and incubation for 30 min at 4 °C with anti-FoxP3 antibody. The lymphocyte subpopulations were determined by gating total lymphocytes based in their FSC/SSC profile, and selecting CD45+ cells; T lymphocyte subpopulations by gating CD45+/CD3+ cells; and T reg cells by gating CD3+/CD4+ cells. The percentage of T-, B-, NK-, and NKT-lymphocytes was determined with respect to total lymphocytes. The percentage of CD4+T lymphocytes and CD8+T lymphocytes was determined with respect to CD3+T lymphocytes, and the percentage of Treg cells with respect to CD4+T lymphocytes. Cells were analyzed on a FACSCanto cytometer (BD Bioscience, Franklin Lanes, NJ, USA). Each sample contained at least 5 × 10^4^ cells and was analyzed with CellQuest-Pro software.

### 4.13. Assays of Immune Protection

Next, 6.25 × 10^5^ cells of Fhit-transfected B9 or B11 tumor cells were subcutaneously inoculated into the footpad in groups of 10 mice. All assays were repeated twice. Thirty days after inoculation, they were subcutaneously inoculated into the other footpad with 6.25 × 10^5^ cells of B9, B11, B7, A7, GR9, CT-26, or 4T1 murine tumor cells. The growth of local tumors was recorded three times/week in all animals, measuring the largest diameter of each tumor with electronic calipers.

### 4.14. Immunohistochemistry Analysis of HLA-I and FHIT Expression in Human Breast Cancers

Frozen tumor samples from primary breast tumors and autologous normal breast tissue samples were obtained from Virgen de las Nieves University Hospital Biobank. Four- to eight-micrometer-thick cryopreserved tumor tissue sections were allowed to dry at room temperature for 4–18 h, fixed in acetone at 4 °C for 10 min, and stored at −40 °C until immunohistological analysis using Biotin-Streptavidin System (NovolinkTM Polymer Detection System, Leica, Newcastle, UK). Sections were counterstained with Mayer’s hematoxylin. Tissues were then incubated for 45 min at room temperature with the primary antibody of interest, including W6/32 (ATCC) and rabbit anti-FHIT (Invitrogen 71–9000). Two pathologists independently evaluated and interpreted the results of immunohistochemical staining, blinded to clinical data of the patients. The staining intensity was graded on a 0–3 scale as follows: 0, absence of staining; 1, weakly stained; 2, moderately stained; and 3, strongly stained. The percentage of positive tumor cells was scored as follows: 0, < 25% of tumor cells; 1, 26–50% of tumor cells; 2, 51–75% of tumor cells; 3, > 75% of tumor cells. For all antibodies used, scoring accounted for both representation of the areas and intensities of the stains. The IHC score (0–9) was calculated by multiplying the intensity and the percentage scores. The total values of 0-1 were scored as negative (−), values of 2–3 as weakly positive (+), values of 4–6 as moderate positive (++), and values of 7–9 as intense positive (+++).

### 4.15. Statistical Analysis

Prism Software (Graph Pad Software V5.0) and SPSS Statistical 25.0 (IBM SPSS Inc., Chicago, IL, USA) were used for statistical analyses. A two-tailed unpaired Student’s *t* test or Mann–Whitney *U* test was used for statistical comparisons between two groups. ANOVA followed by the Tukey post-hoc analysis or Kruskal–Wallis test followed by the Dunn’s post-test was used for multiple comparisons. Fisher exact test was used to analyzed the interrelation between two qualitative variables. Spearman rank correlation was used to analyze the relationship between two variables. All data are expressed as means ± standard deviation (SD) or means ± SEM. *p <* 0.05 was considered statistically significant (* *p* < 0.05, ** *p* < 0.01, *** *p* < 0.001).

## 5. Conclusions

Our findings show that Fhit regulates MHC-I expression on tumor cells. The restoration of MHC-I expression on tumor cells by Fhit gene transfection generates a strong immune response mediated by T-lymphocytes—mainly cytotoxic T-lymphocytes—producing the immune rejection of highly oncogenic tumor cells. Strikingly, the pattern of immune rejection differed between the sexes, being observed in all of the females and around half of the males. Moreover, the immune response generated in the hosts was protective against the same (non-transfected) tumor cells and other tumor cells in the GR9 model. We showed a high significant association between FHIT protein expression loss and downregulation of HLA class I molecules on human breast tumors. According to these findings, FHIT expression induction and the consequent recovery of MHC-I expression on tumor cells suggests a potential immunotherapeutic strategy. Further research is warranted to explore the possibility that FHIT expression recovery by in vivo FHIT-transfection or other means, or the use of in vitro irradiated Fhit-transfected tumor cells might be useful as a personalized immunotherapeutic antitumor vaccine in human patients with FHIT-negative and MHC-I-downregulated tumors. Unlike the present strategy, cancer vaccines or immunotherapies based on tumor antigens, dendritic cells, or immune-checkpoint antibodies require positive MHC-I surface expression on tumor cells to be effective [54]. For instance, the sole FDA-approved cancer vaccine is only effective when prostate tumor cells express MHC-I [55]. Hence, our anticancer vaccine may be a useful approach before the initiation of other immunotherapies and may even act as a personalized anti-tumor immunotherapeutic vaccine.

## Figures and Tables

**Figure 1 cancers-12-01563-f001:**
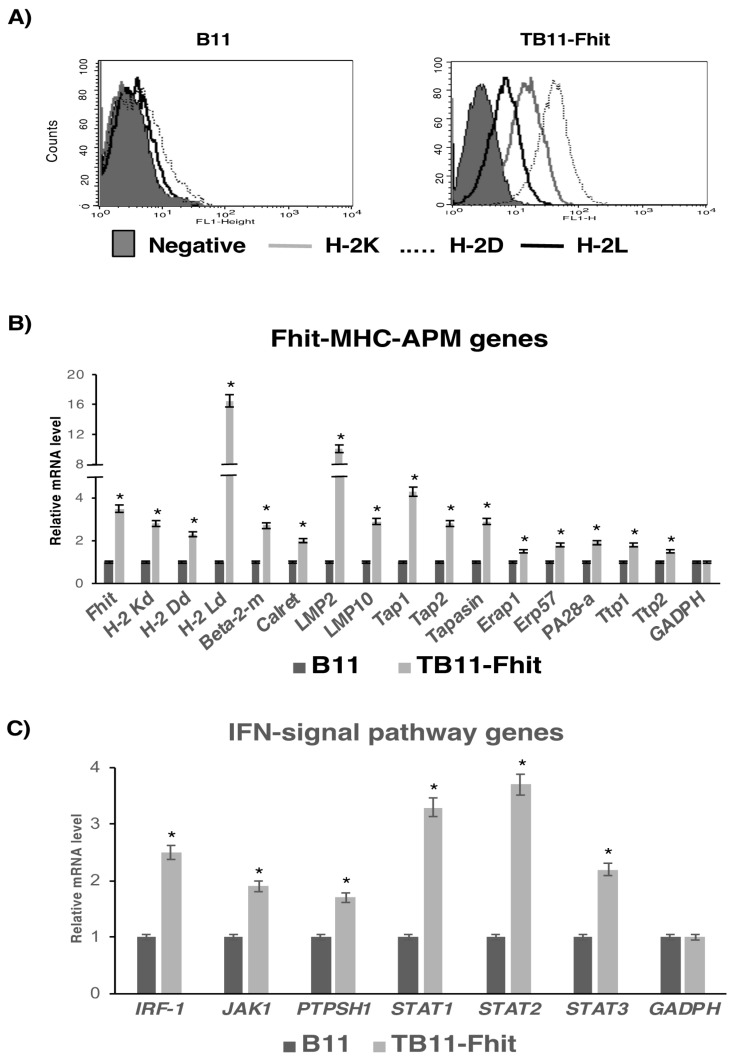
Changes produced in B11 tumor cells by the Fhit gene transfection. (**A**) MHC class I surface expression of B11 and TB11-Fhit tumor cell lines in baseline conditions: H-2 K^d^ (gray line), H-2 D^d^ (dotted line), and H-2 L^d^ (black line). B11 is negative in baseline conditions; TB11-Fhit recovered surface expression of all three H-2 molecules. Data from one experiment are depicted; (**B**) Transcription levels of Fhit, H-2 class I heavy chain, β_2_microglobulin, and several APM components detected by real-time RT-PCR; (**C**) Transcriptional expression of IFN-γ signal transduction pathway genes in B11 and TB11-Fhit tumor cell lines. Data were normalized using β-actin and GAPDH as housekeeping genes. Only genes with changes in their expression are depicted. Data for B11 are set at 1. Values are depicted as means ± SD of three independent experiments performed in quadruplicate. * *p* < 0.05. A two-tailed Student’s *t*-test was used for statistical analysis.

**Figure 2 cancers-12-01563-f002:**
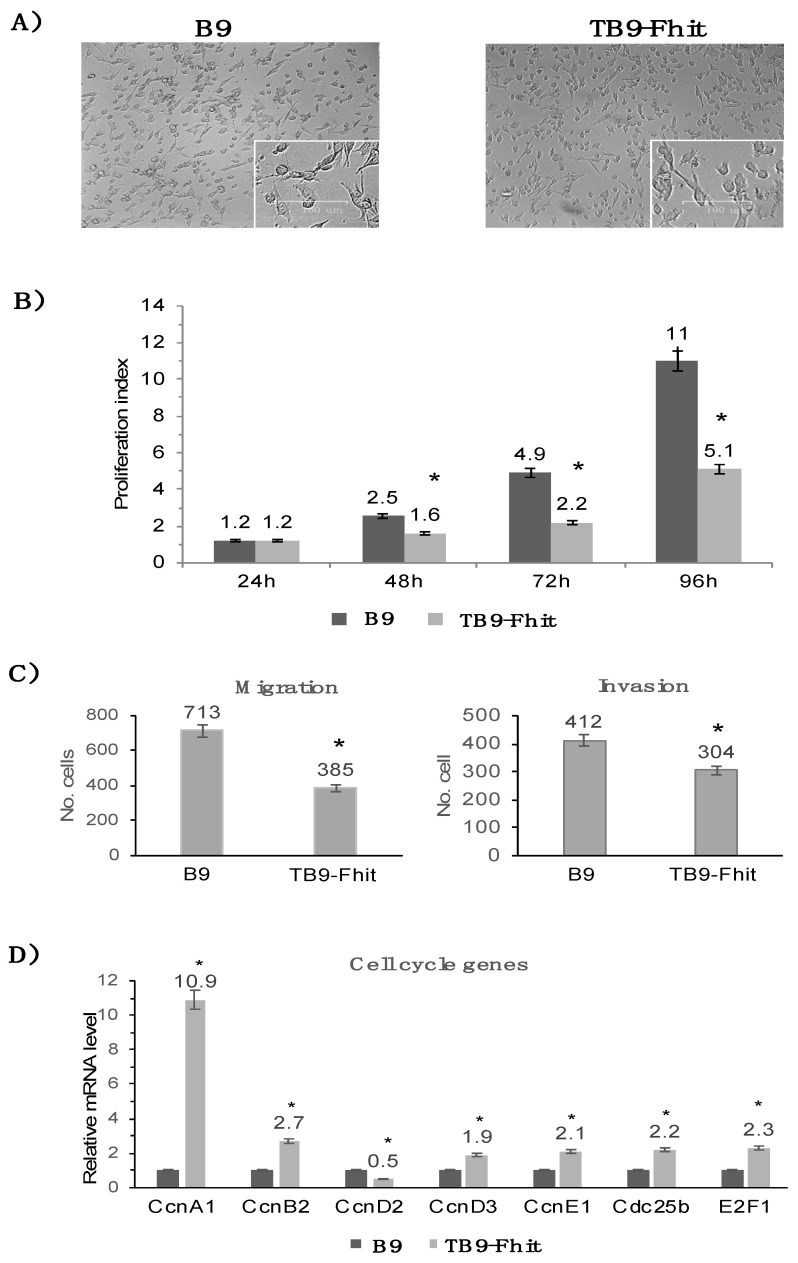
In vitro changes produced in B9 tumor cells by the Fhit gene transfection. (**A**) Microscopic image of B9 and TB9-Fhit tumor cell lines. Both tumor cell lines show very similar cellular morphology; (**B**) In vitro proliferation index of B9 and TB9-Fhit tumor cell lines. TB9-Fhit shows a lower proliferation index. The proliferation index was calculated: final cell number/initial cell number. Values are depicted as means ± SD of three independent experiments performed; (**C**) In vitro migration and invasion assays compare B9 and TB9-Fhit tumor cells. TB9-Fhit shows lower migratory and invasive potential. Values are depicted as means ± SD of three independent experiments; (**D**) Transcriptional expression of cell cycle genes in B9 and TB9-Fhit tumor cell lines. Data were normalized using β-actin and GAPDH as housekeeping genes. Only genes with changes in their expression are depicted. Data for B9 are set at 1. Values are depicted as means ± SD of three independent experiments performed in quadruplicate. * *p* < 0.05. A two-tailed Student’s *t*-test was used for statistical analysis. Scale bar, 100 μm.

**Figure 3 cancers-12-01563-f003:**
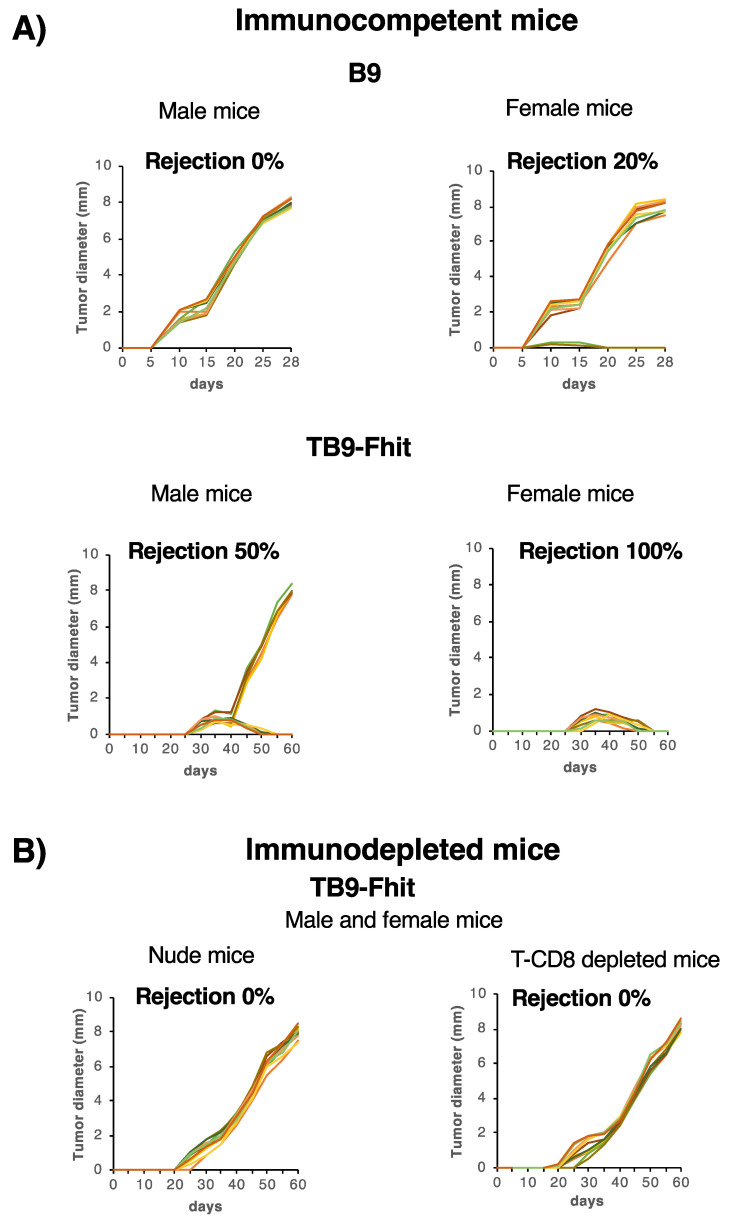
In vivo oncogenicity of untransfected and Fhit-transfected tumor cells in immunocompetent and immunodepleted mice. (**A**) In vivo tumor growth curves (*n* = 10 mice per group) of B9 and TB9-Fhit tumor cells (cell dose 6.25 × 10^5^) in female/male immunocompetent mice. TB9-Fhit was rejected in 100% of female mice and 50% of male mice. Fisher’s exact test showed that tumor rejection significantly differed between male and female mice. Assays were repeated twice; (**B**) In vivo tumor growth curves (*n* = 10 mice per group) of TB9-Fhit tumor cells (cell dose 6.25 × 10^5^) in female nude mice. Identical results were found in male nude mice and in CD8+ T lymphocyte-immunodepleted male/female immunocompetent mice. TB9-Fhit tumor cells grew in all animals. Assays were repeated twice.

**Figure 4 cancers-12-01563-f004:**
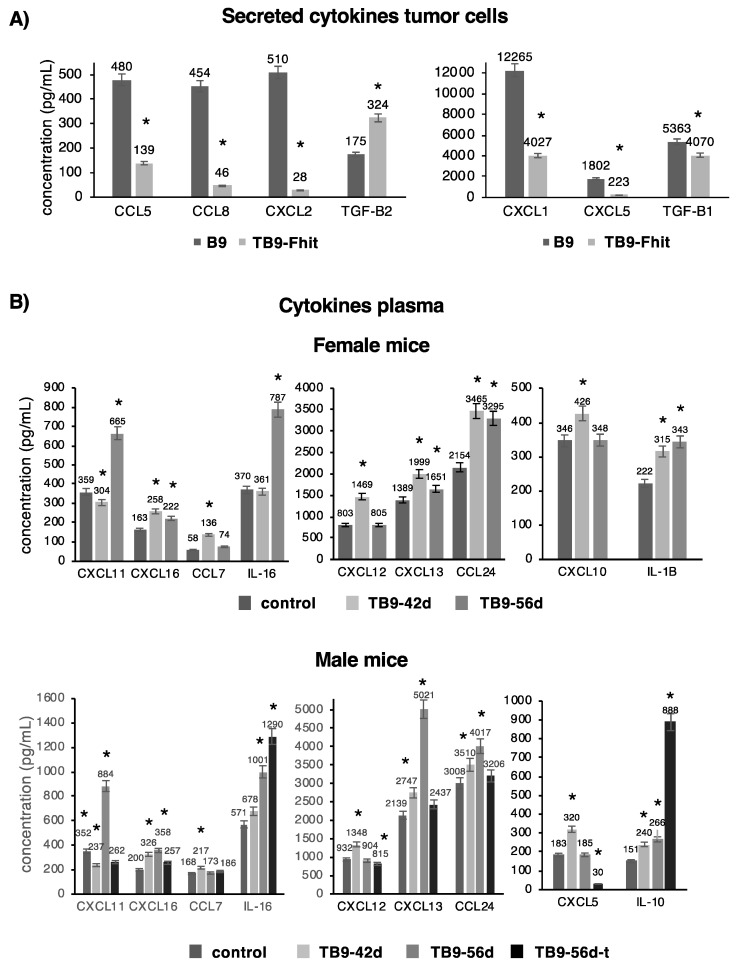
The cytokine/chemokine profile in untransfected and Fhit-transfected tumor cells and in mice inoculated with Fhit-transfected tumor cells. (**A**) The cytokine/chemokine profile secreted by Fhit-transfected versus non-transfected tumor cells. Fhit transfection of tumor cells modified the pattern of cytokine/chemokine secretions; (**B**) The cytokine/chemokine profile in plasma of female/male mice inoculated with Fhit-transfected tumor cells versus control mice. Results show chemokine/cytokine levels at 42 (42d) and 56 (56d) days post-inoculation of tumor cells. The male mice with primary tumor are depicted as 56d-t. Only cytokines/chemokines presented changes in their expression are depicted. Differences were found between female and male mice and between male mice with and without a primary tumor. Values are depicted as means ± SD of two independent experiments performed in duplicate. * *p* < 0.05. A two-tailed Student’s *t*-test or ANOVA test, followed by Tukey’s post-hoc test, was used for statistical analysis.

**Figure 5 cancers-12-01563-f005:**
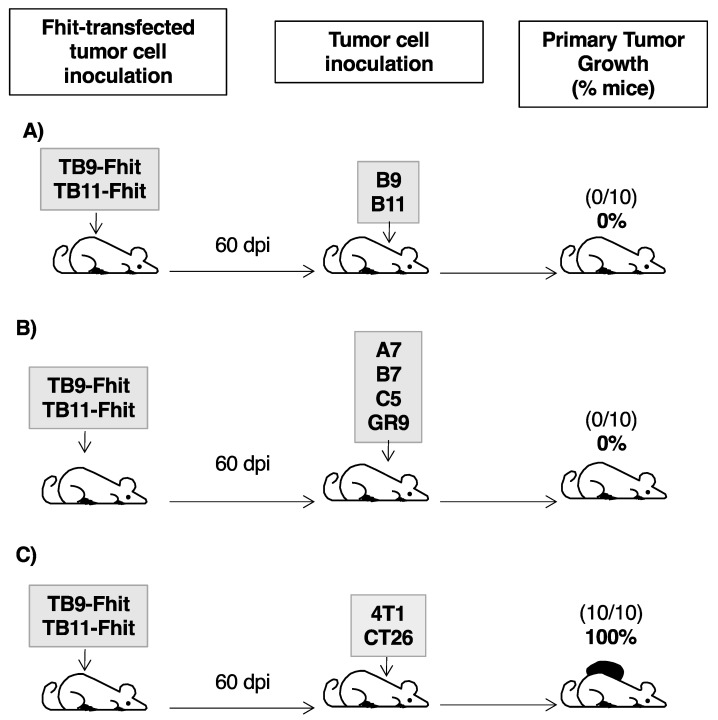
Immunogenicity and immunoprotection of female mice inoculated with Fhit-transfected B9/B11 tumor cells. (**A**) TB9- and TB11-Fhit immunoprotected the hosts against B9 and B11 tumor cells. The animals (*n* = 10 mice per group) were previously inoculated with Fhit-transfected tumor cells and then, 60 days later, inoculated with non-transfected tumor cells; B9 and B11 tumor cells were immune rejected in all mice; (**B**) TB9- and TB11-Fhit immunoprotected the hosts against other tumor cell clones of the GR9 tumor system and even against GR9 bulk tumor cells; (**C**) TB9- and TB11-Fhit did not immunoprotect the hosts against 4T1 breast carcinoma cells or CT26 colon carcinoma cells. The assays were repeated twice.

**Figure 6 cancers-12-01563-f006:**
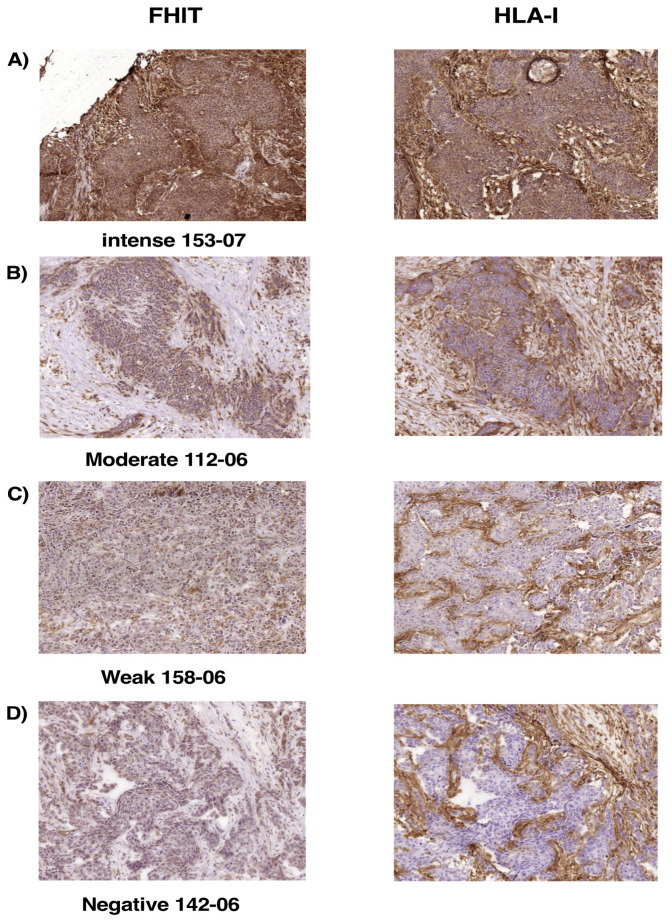
Expression of FHIT and MHC-I in human breast carcinoma cells analyzed by immunohistochemistry. (**A**) Representative tumor with intense FHIT and HLA-I expressions; (**B**) Representative tumor with moderate FHIT and HLA-I expressions; (**C**) Representative tumor with weak FHIT and HLA-I expressions. (**D**) Representative tumor with negative FHIT and HLA-I expressions. The same tumor cells were analyzed in each case for FHIT and HLA-I expression. Magnification 200×.

**Table 1 cancers-12-01563-t001:** Changes in splenic leukocyte populations.

**Splenic Leukocyte Populations**							
**Females**	**CD3^+^**	**CD3^+^** **CD8^+^**	**CD3^+^** **CD4^+^**	**CD3^+^** **CD49b^+^**	**CD3^−^** **CD19^+^**	**CD3^−^** **CD49b^+^**	**^a^ CD4^+^ CD25^+^** **FoxP3^+^**
**Control**	51.6 ± 1.2	21.3 ± 1.0	30.4 ± 0.7	3.6 ± 0.4	42.5 ± 1.0	5.9 ± 0.4	5.5 ± 0.4
**14 dpi ^d^**	43.6 ± 2.7 ^c^	17.3 ± 1.7 ^b^	26.2 ± 2.9 ^b^	3.4 ± 0.3	51.5 ± 2.0 ^c^	4.7 ± 0.7	5.9 ± 0.7
**21 dpi**	56.6 ± 3.9 ^b^	27.0 ± 2.3 ^c^	29.6 ± 3.5	4.6 ± 0.5	37.0 ± 4.2 ^b^	5.4 ± 0.4	6.87 ± 0.5
**28 dpi**	61.0 ± 3.2 ^c^	26.6 ± 1.8 ^c^	34.3 ± 2.5 ^b^	5.0 ± 0.2 ^b^	32.0 ± 2.9 ^c^	7.1 ± 1.0	4.7 ± 0.4
**Splenic Leukocyte Populations**							
**Males**	**CD3^+^**	**CD3^+^** **CD8^+^**	**CD3^+^** **CD4^+^**	**CD3^+^** **CD49b^+^**	**CD3^−^** **CD19^+^**	**CD3^−^** **CD49b^+^**	**^a^ CD4^+^ CD25^+^** **FoxP3**
**Control**	44.0 ± 0.7	17.8 ± 1.2	26.3 ± 1.8	2.9 ± 0.3	47.7 ± 0.8	7.7 ± 1.2	7.3 ± 0.8
**14 dpi**	44.1 ± 1.1	16.9 ± 0.4	27.2 ± 1.2	2.5 ± 0.2	51.6 ± 0.8 ^b^	4.3 ± 0.4 ^b^	6.5 ± 0.1
**21 dpi**	45.0 ± 2.2	19.7 ± 0.8	25.3 ± 1.0	3.2 ± 0.3	50.2 ± 2.1	4.8 ± 0.2 ^b^	6.1 ± 0.3
**28 dpi**	49.4 ± 1.8 ^c^	20.9 ± 1.3 ^b^	28.5 ± 2.5	4.0 ± 0.3	44.7 ± 1.4 ^b^	5.7 ± 0.5	5.4 ± 0.5 ^b^

(*n* = 8 mice per group; mean ± SEM). All assays were repeated twice. ^a^ Percentage among CD4+ cells. ^b^
*p* < 0.05 compared with control group. ^c^
*p* < 0.01 compared with control group. ^d^ Days post injection of the TB9-Fhit tumor cells.

**Table 2 cancers-12-01563-t002:** FHIT and MHC-I expression in breast cancers.

Tumor Sample	Score ^a^	Expression ^b^
Breast Cancer	FHIT/HLA-I	FHIT/HLA-I
27–05	9/9	+++/+++
44–07	9/9	+++/+++
69–07	9/9	+++/+++
75–05	9/9	+++/+++
94–06	9/9	+++/+++
105–05	9/9	+++/+++
119–05	9/9	+++/+++
120–04	9/9	+++/+++
120–06	9/9	+++/+++
123–07	9/9	+++/+++
153–07	9/9	+++/+++
163–07	9/9	+++/+++
192–06	9/9	+++/+++
201–06	9/9	+++/+++
250–07	9/9	+++/+++
148–04	9/6	+++/++
205–06	6/9	++/+++
4–07	6/6	++/++
35–07	6/6	++/++
112–06	6/6	++/++
133–05	4/4	++/++
32–07	4/0	++/-
56–06	3/3	+/+
93–07	3/3	+/+
191–05	3/3	+/+
196–06	3/3	+/+
203–06	3/3	+/+
95–06	3/0	+/−
207–06	3/0	+/−
41–07	2/2	+/+
125–07	2/2	+/+
128–06	2/2	+/+
158–06	2/2	+/+
118–06	2/0	+/−
136–06	2/0	+/−
137–06	2/0	+/−
203–05	2/0	+/−
16–07	0/0	−/−
23–07	0/0	−/−
100–06	0/0	−/−
108–06	0/0	−/−
120–07	0/0	−/−
134–06	0/0	−/−
142–06	0/0	−/−
150–07	0/0	−/−
154–05	0/0	−/−
186–07	0/0	−/−
204–06	0/0	−/−

^a^ Final score was calculated by multiplying intensity and extent of stained cells. ^b^ Expression was determined according to score: +++, 7–9; ++, 4–6; +, 2–3; −, 0–1. Rs = 0.925, *p* < 0.001. Spearman’s rank-correlation test was used for statistical analysis.

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
