# Peer review of "Restoration of MHC-I on Tumor Cells by Fhit Transfection Promotes Immune Rejection and Acts as an Individualized Immunotherapeutic Vaccine"

_cancers, 2020, doi:10.3390/cancers12061563_

Round 1

Reviewer 1 Report

Title: Restoration of MHC-I on tumor cells by Fit transfection promotes immune rejection and acts as an individualized immunotherapeutic vaccine

In present manuscript the show that Fhit gene regulates MHC-I cell surface expression on tumor cells. The restoration of MHC-I expression on tumor cells by Fhit gene transfection generates a strong immune response mainly mediated by cytotoxic T-lymphocytes, producing the immune rejection of highly oncogenic tumor cells.

The topic of the present study is up to date and very relevant and the paper responds well to the objectives of the study. Recovery of Fhit expression on MHC class I negative tumor cells may be a useful immunotherapeutic strategy and may even act as an individualized immunotherapeutic vaccine.

Overall, data and experimental design are very effective to describe the hypothesis of the authors.

There is only one thing that the authors should explain better. They assert that the Fhit-transfected tumor cells acted as an immunotherapeutic antitumor vaccine, because “all female mice injected with these cells gained immune protection against the progression of cancer generated from the same (non-transfected) tumor cells, from another tumor clone derived from GR9 tumor model system, and even from GR9 bulk tumor cells”. But is not clear how the cytotoxic immune response triggered by the Fhit-transfected tumor cells may recognize and destroy the non-transfected one. In fact, the non-transfected cells do not express or express on their surface very few antigens having low expression of the MHC class I complex.

Reviewer 2 Report

The manuscript by Pulido et al, showed the effect of transfection of Fhit gene in another (B11 Clone) MHC-I-negative tumor clone of the GR9 mouse tumor model, in addition the author examines the effect of Fhit transfection and MHC-I cell surface expression restoration on the major cell activities. The title does not reflect incisively the results obtained.They seem to be several experiments put together without a real common thread. Many experiments presented show results that are not illustrated in the right way.

Reviewer 3 Report

General issues: 

In the experiment depicted in Fig. 5 the authors show that pre-exposure to an Fhit-transfected tumor line renders mice resistant to tumor formation by the non-transfected line. This is very interesting, because it points to a potential way for therapeutic intervention (see below). It is also a puzzling result because in the case of B9, the tumor cells are basically MHC I negative (Fig. S4)The authors have shown in an earlier experiment that survival of exposure to Fhit-transfected tumor lines depend on a CTL-response (Fig. 3). So the question is, what kind of immune response is induced by the Fhit-transfected cells to kill the non-transfected tumor line later? Is it CTL-dependent scanning of virtually non-existent MHC I-peptide complexes on the cell surface? The authors mention in the discussion that some tumor lines re-isolated from mice regained MHC I expression to some extent. Or are antibody-dependent responses required? After all, the authors have shown that at least some expansion of B-cells occurs in their experimental conditions. The authors may address these questions by depleting CD8- or CD4-T-cells from mice previously primed with Fhit-transfected B9/B11-tumor lines. A subsequent challenge with untransfected B9/B11 cells should then reveal which type of immune response is required for clearing the untransfected tumor cells. 

Another question arising from their experimental results is whether Fhit-transfected tumor lines could be used to immunize against the parent tumor at a stage where the parent tumor is already established. This would be a question more relevant to a potential clinical application of the research presented in this manuscript. Tackling such a question successfully would certainly improve the impact of the study. 

Specific issues: 

  1. In Fig.1 the authors should add Western blot data on the expression of APMgene products to substantiate their claim that the antigen processing machinery serving MHC I is indeed upregulated upon Fhit transfection 
  2. In Fig. 2 the lettering of the graphs is very discordant and makes the whole figure to look ugly, please amend. Furthermore, in Fig. 2 A, the cell morphology cannot be judged reasonably. Please enlarge smaller areas of both micrographs! It is also considered standard to include size bars in micrographs. 
  3. In Fig. 3 B data on nude mice and CD8 T-cell immunodepleted mice should be shown in separate graphs. Furthermore, the authors mention data on CD4 T-cell depleted mice that should be included here as a control. When necessary, separate graphs on male and female recipients should be shown. The lines in all graphs of Fig. 3 should be plotted thinner and perhaps could be colored to make the individual developments of tumor diameters discernible. 
  4. In Fig. 4 the selection of different cytokines and different parameters in the diagrams of female and male mice makes it difficult for the reader to compare the data. It is not evident for the reviewer why different cytokines/chemokines were analyzed in the two groups. Please add the data on the missing ones and arrange all data in such a way that they can be compared straightforwardly. 
  5. In the text covering Fig. 5 the authors do not mention that they obviously have used female mice for the experiments depicted there. Funny enough, this piece of information is provided in the discussion! Please move to the appropriate place! 
  6. In line 255 the authors overinterpret a series of experiments by using the term “individualized immunotherapeutic vaccine”. This is a bit embarrassing since for clinical relevance a kind of vaccination against an already manifest tumor would be needed, wouldn’t it. The authors, however, may address this issue, as suggested above in the general section. 
  7. In line 115 the authors mention that Fhit gene expression was not induced by IFN-gamma. Since this is an important piece of information, they should either show the data or cite it properly. 
  8. Line 216: the authors refer to supplementary table 1, but the information is in suppl. table 2. 

Reviewer 4 Report

In this paper authors followed on their previous finding that demonstrate MHC-I restoration by the ectopic expression of the Fhit, related to antigen processing and presentation machinery in mouse models of fibrosacoma that showed anti-tumor immune rejections. Here, they demonstrate that expression of Fhit in MHC-I negative tumor cell lines was associated with a decrease in cancer cell proliferation, invasion and migration in vitro and tumors rejection in female mice. While no mechanism was reported on gender-dependency of anti-tumor immune response in this model, they show some interesting data on the potential Fhit-dependent vaccination. This vaccination works only for one model of murine mammary tumor GR9 but not in the classical model of 4T1 (with the genetic background) or colorectal mouse model of CT26. Thus, authors conclude that Fhit can be used as individualized immunotherapeutic vaccine. This was supported by clinical data on human breast cancer samples in which they find that at least 45 % of these samples have an association between FHIT and HLA-1 expression.

Comments

  • In there previous report (Frontiers Immunology 2018), they revealed that high MHC-I expression was associated with low local oncogenicity and high spontaneous metastatic capacity, whereas MHC-I-low clones showed high local oncogenicity and no spontaneous metastatic capacity. What is the effect of Fhit expression on metastasis in males and females mice, related to tumor growth data reported in Figure 3?
  • How to explain the effect of Fhit expression on cancer cell migration, invasion and proliferation decrease in vitro?
  • What are the levels of other markers of immune response such as TIM3, PD-1, PD-L1, CD137L, CD137, Fas, and FasL in tumor in male and in female mice?
  • Tumor immune response seen in control females may be due to their basal levels of T cells, T helper and CTL that seem to be higher than that of control males (table 1:CD3+, 51.6 vs 44; CD8, 21.3 vs 17.8; CD3+CD4+, 30,4 vs 26.3). This should be discussed as it might alter the take home message of the study and the real effect of Fhit.
  • Is there any clinicopathological features of human samples (table 2) positive for Fhit ?
  • Figure 5a, data should be presented here in addition to the schematic illustration of the finding.
  • Figure 3b, Data of CD8+ T lymphocyte depletion should be presented here.
  • Line 206: “in the former (Table 1)” the use of former here is confusing, please correct.

Reviewer 5 Report

The authors recover MHC-I expression by Fhit gene transfection in the MHC-I negative tumor cell lines and analyze their oncological and immunological profile. They show that Fhit-transfected tumor cells possess high immunogenic, and this T cell-mediated immune rejection is more significant in females than that in males. In addition, they also observe a direct correlation between FHIT and HLA-I expressions in human breast tumors. Thus, they suggest that recovery of Fhit expression on MHC class I-negative tumor cells may be a useful immunotherapeutic strategy.

This study is interesting; however, the data are mostly observations and lack mechanism. In addition, the proposal to use Fhit-transfected tumor cell as vaccine is dangerous. The English writing should be improved. Thus, I suggest that this manuscript should be rejected.

Comments

  1. The authors should add introduction about the background of FHIT.
  2. The “MHC-I cell surface expression” should be “MHC-I expression” in all text.
  3. They have to describe the difference between B9 and B11 tumor cells.
  4. Legends of Figs 1-5 should have titles.
  5. Line 185 in p.6, Fig 3B does not show the data for CD4 and CD8 depletion.
  6. In Fig 6, they only compare the expressions of Fhit and HLA-I. They should compare the Fhit expression to the survival rate.
  7. It is dangerous to immunize tumor cell as vaccine. They should propose how to deliver Fhit into tumor cell in vivo, or use dead tumor cells. 

Round 2

Reviewer 3 Report

The authors have convincingly addressed the points raised by the reviewer.

Author Response

We are grateful to the reviewer for their helpful comments, which proved invaluable in preparing this improved version

Reviewer 4 Report

The revised manuscript was improved and the reviewer's comment were correctly addressed. 

Author Response

(The authors gave the same response as above.)

Reviewer 5 Report

Point 1: however, the data are mostly observations and lack mechanism Response 1: We refute this comment. Data presented in this study reveal: a) an increase in APM, IFN pathway genes and MHC-I expression after Fhit transfection; b) determination of the leukocyte subpopulations and cytokines/chemokines involved in the rejection of Fhit transfected tumor cells; c) the usefulness of Fhit transfection as immunotherapeutic vaccine; d) direct association between FHIT and HLA-I expression in human breast cancer; among others. All of these results provide the mechanisms for our observations.

Is Fhit a transcriptional factor? How can Fhit increase these gene expression? They should have a discussion, at least.

Point 3: The authors should add introduction about the background of FHIT.
Response 3: We think that the introduction should be focused on the MHC-I molecules and
the immune system, as well as on the clear description of our tumor model. The background
of FHIT is commented in the discussion section according to our results (lines 359-368).

Fhit is your target.
